# Psychobiotic Effects on Anxiety Are Modulated by Lifestyle Behaviors: A Randomized Placebo-Controlled Trial on Healthy Adults

**DOI:** 10.3390/nu15071706

**Published:** 2023-03-31

**Authors:** Ricardo Morales-Torres, Cristóbal Carrasco-Gubernatis, Aitana Grasso-Cladera, Diego Cosmelli, Francisco J. Parada, Ismael Palacios-García

**Affiliations:** 1Center for Cognitive Neuroscience, Duke University, Durham, NC 27710, USA; rim11@duke.edu; 2Centro de Estudios en Neurociencia Humana y Neuropsicología, Facultad de Psicología, Universidad Diego Portales, Santiago 8320000, Chile; cristobal.carrasco1@mail.udp.cl (C.C.-G.);; 3Programa de Magíster en Neurociencia Social, Facultad de Psicología, Universidad Diego Portales, Santiago 8320000, Chile; 4Escuela de Psicología, Facultad de Ciencias Sociales y Centro Interdisciplinario de Neurociencias, Pontificia Universidad Católica de Chile, Santiago 7820436, Chile

**Keywords:** microbiota-gut-brain axis, psychobiotics, mental health, lifestyle behaviors, well-being, Mediterranean diet, embodied mind

## Abstract

Psychobiotics are modulators of the Microbiota-Gut-Brain Axis (MGBA) with promising benefits to mental health. Lifestyle behaviors are established modulators of both mental health and the MGBA. This randomized placebo-controlled clinical trial (NCT04823533) on healthy adults (*N* = 135) tested 4 weeks of probiotic supplementation (*Lactobacillus helveticus* R0052 and *Bifidobacterium longum* R0175). We assessed effects on wellbeing, quality of life, emotional regulation, anxiety, mindfulness and interoceptive awareness. We then analyzed if lifestyle behaviors modulated probiotic effectiveness. Results showed no significant effects of probiotic intake in whole sample outcomes. Correlational analyses revealed Healthy Behaviors were significantly correlated with wellbeing across scales. Moreover, the linear mixed-effects model showed that the interaction between high scores in Healthy Behaviors and probiotic intake was the single significant predictor of positive effects on anxiety, emotional regulation, and mindfulness in post-treatment outcomes. These findings highlight the relevance of controlling for lifestyle behaviors in psychobiotic and mental health research.

## 1. Introduction

Communication between extracranial signals and the brain has received increasing attention in recent decades given its impact on cognition and mental processes [1,2,3]. The gut-brain axis is one of the most relevant topics of this body-brain research, positioning the gut microbiota as a target for clinical interest [4]. The microbiota-gut-brain axis (MGBA) refers to the complex communication system between the human gastrointestinal tract, the micro-organisms which inhabit it, and the peripheral and central nervous systems [5]. Various mechanisms are implicated in such axis, including a strong immunological regulation, neural signaling through the autonomic nervous system (primarily via cranial nerve X, vagus nerve), the production of microbial metabolites (e.g., bile acids, choline, and short-chain fatty acids), and metabolism of neurotransmitters (e.g., GABA, noradrenaline, serotonin, dopamine, acetylcholine) [5]. As a result of this, the study of the MGBA has begun to reveal different ways in which they can be implicated in human health, from psychopathological conditions [6] to well-being [7]. In fact, human trials have linked the microbiome with specific processes that are expected to contribute to well-being, such as emotional regulation [8], motivation [9], stress and anxiety [10], and vitality [11], among others. However, the extent to which the MGBA dynamics affect well-being as an independent construct and the psychological processes beyond that may contribute to it have not yet been explored.

Well-being can be defined as the optimal psychological functioning and its experience, where physical and subjective factors are integrated [12]. Recently, a theoretical framework was developed in order to include the microbiome in the current debates of cognitive science, giving theoretical support to future research agendas [2,13]. In brief, human psychological processes and functioning might depend in part on: (i) the brain, which is inseparable from a body, with the microbiome being a functional part of this body, and (ii) behavioral activities developed through people’s lifestyles (i.e., diet, physical activity, contact with nature), which in turn are some of the greatest adult human microbiome modulators. Thus, it could be hypothesized that microbiome modifications should impact psychological functioning and well-being in a way that may be modulated by people’s lifestyle.

One of the most studied interventions that modulate microbiome physiology and possibly affect human behavior is psychobiotics administration [1,4]. In particular, it has been shown that a mixture of *Lactobacillus* and *Bifidobacterium* strains exerts some beneficial effects in healthy populations, as measured by different scales accounting for anxiety, stress, and psychopathological disorders [6,14]. Recent meta-analyses have shown a little overall effect on several pooled outcomes regarding mental health [1,6,15,16], concluding that an optimal psychobiotic recommendation in terms of dosage, strains, and duration cannot yet be conceived. As hypothesized, lifestyle patterns may be considered an important aspect at the moment of designing psychobiotics interventions, especially given that factors such as diet or environmental contact are crucial microbiome modulators during adulthood [17,18]. For instance, different probiotic effectiveness should be expected in people with different nutritional patterns (e.g., Mediterranean vs. Western) or different levels of physical activity.

In this randomized controlled trial, we aimed to assess the effect of a psychobiotic formulation specifically on well-being, exploring the extent to which these effects might be modulated by lifestyle factors such as diet, physical activity, social connectedness, among others.

## 2. Materials and Methods

### 2.1. Experimental Design and Procedures

This was a four week, double-blind, randomized, placebo-controlled trial (ClinicalTrials.gov NCT04823533). The final experimental design is shown in Figure 1. Initially, the study used a factorial design (2 × 2) with a probiotic/placebo intake and a gastric interoceptive intervention in healthy people from Santiago, Chile. However, difficulties attributed to the pandemic did not allow us to implement the interoceptive intervention properly, therefore it was eliminated from the protocol. After assessing our eligibility criteria, participants were randomized into one of the two experimental groups: (1) Probiotic intake; (2) Placebo intake (Figure 1). All participants completed a self-report survey with eight self-reported and validated scales, completed a custom-made lifestyle survey, and delivered a stool sample (in the present report) at three different time points: at the beginning of the experiment, before any intervention (‘Baseline’ measurements), after the 4-week intervention (‘Post’ measurement) and after eight weeks since the ‘post’ measurements, and without any instruction (‘Follow-up’ measurement).

Prior to the beginning of the experiment, all participants received a bag of materials. Each bag contained detailed protocol instructions, materials required for the three measurements of stool sampling and storage, and the probiotic or placebo sticks for the full experiment. Participants filled out the questionnaires and collected the stool sample on the 3 study timepoints, at days 1, 28, and 84. Following a very strict protocol of hygienic sample collection, participants stored samples at −20 °C in the freezer section of their home refrigerator until they were picked up by the researcher the same day. In the present report, no additional information regarding the stool sampling and processing will be delivered, given that no reference to this data nor results will be done.

This study obtained approval of the Bioethics Committee of the Faculty of Medicine at the Pontificia Universidad Católica de Chile. Written informed consent was obtained from all participants before the study. Reporting of findings was done in accordance with the Consolidated Standards of Reporting Trials (CONSORT) guidelines [19].

### 2.2. Participants

#### 2.2.1. Recruitments Strategies

Participants were recruited in Santiago de Chile through an open call on social media (i.e., Instagram) from January 2021 to February 2021. Social media-based recruitment strategies were used to invite participants with video posts explaining the experiment and main outcomes (brain-gut-microbiota axis health and well-being). Interested people fulfilled an anonymous self-report survey giving the information necessary to assess the inclusion/exclusion criteria. Selected candidates received an email with the online version of the informed consent and general information about the intervention, sampling, and timelines. Interested candidates confirmed their participation via email and signed the informed consent prior to initiation of the experimental procedures.

#### 2.2.2. Inclusion and Exclusion Criteria

Potential participants completed an anonymous self-report survey to evaluate eligibility. Inclusion criteria were evaluated at screening and included: being aged between 18–65 years, living in Santiago de Chile, successfully completing the self-report survey (via Google Forms), being able to provide written consent, and willing to comply with study requirements, namely providing stool samples, completing questionnaires and records, and discontinuing consumption of any other probiotic supplement for the four weeks previous the experiment and for the duration of the study. Exclusion criteria included: being pregnant or planning to get pregnant at the time of enrollment, breastfeeding, being infected with SARS-CoV-2, consuming probiotics, having used antibiotics during the last month before enrollment, having been diagnosed with one of the following: diabetes or metabolic diseases in the last 5 years, depression, psychiatric or neurological diagnostic in the last year, IBS, gastrointestinal diseases or received bariatric surgery in the last 5 years, immune disorders or possible immune-deficient status in the last 5 years, or with milk or soy allergy.

#### 2.2.3. Randomization and Blinding

Randomization sequences were generated using basic excel functions by two independent research assistants sequentially (i.e., one of them generated the sequences, and the other corroborated the process). Both research assistants performed the randomization allocation and prepared the materials needed by the participants. The sensorially identical (color, shape, taste) probiotic or placebo sticks were randomly added to each bag by the independent assistant in a 1:1 ratio. Each bag with a particular number (between 1 and 143) was coded with the specific product code number and each participant was assigned to a random number between 1 and 143. Unblinding was done after the first round of analysis.

#### 2.2.4. Sample Size

Sample size was calculated using the statistical program G*Power [20]. The parameters were defined according to previous studies for the evaluation of the effect of probiotics in well-being scales [21]. Using as a statistical test a repeated measure ANOVA, assuming a small effect size of 0.25 and an alpha value of 0.05, our sample of 134 participants allowed us to detect an effect size of 0.88.

### 2.3. Probiotic Treatment

The probiotic/placebo intake intervention was carried out for four weeks from March to July 2021. The probiotic contains 3 billion CFU of two well-documented probiotic strains (*Lactobacillus helveticus* R00052 and *Bifidobacterium longum* R0175) with excipients (xylitol, maltodextrin, plum flavor, and malic acid), under the commercial name of Cerebiome^®^. Placebo contains only the excipients. Participants were instructed to dissolve the probiotic/placebo stick in a 300 mL glass of water once a day (ideally at the same hour everyday). They were instructed to register the date and intake time for each day.

### 2.4. Variables and Measurements

This study considers different measures at three time points: a baseline measurement (‘Baseline)’, previous to interventions; a post intervention measurement (‘Post’, 4 weeks after baseline), and a follow-up measurement (‘Follow-up’, 12 weeks after baseline). The measurements were conducted by a self-report survey in which participants had to complete eight validated scales, related to well-being and lifestyle habits. Due to sanitary conditions caused by the COVID-19 pandemic, the questionnaires were administered online (via SurveyMonkey Inc., San Mateo, CA, USA). For this study, the Spanish version of each questionnaire was used.

The mental/psychological aspect of well-being has been studied from two main perspectives: the eudaimonic and the hedonic, also called psychological and subjective well-being, respectively [22]. The main approach for studying eudaimonic well-being arises from the model of Ryff [23]. Based on psychological concepts such as self-actualization and optimal development, this conceptualization focuses on answering the question, “what constitutes essential features of well-being?”. On the other hand, the most used approach to measure subjective well-being is the hedonic approach, based on a broad ‘happiness’ notion through emotional components (i.e., positive-negative affect) and a cognitive component considered ‘satisfaction with life’ [24], defined as the overall conception that a person has of their own life [25]. In order to assess well-being, for our primary outcome, we selected the Ryff well-being Scale (RYFF) (α = 0.89) [26]. This scale consists of 29 items that assess six aspects of well-being and happiness (autonomy, environmental mastery, personal growth, positive relations with others, purpose in life, and self-acceptance).

Given the well-known multidimensionality of well-being and its close relation to concepts such as quality of life, we also included three secondary measures of well-being: (a) Positive And Negative Affect Scales (PANAS) [27], a 20-item questionnaire used to assess mood by two scales (Positive Affect subscale: α = 0.78; Negative Affect subscale: α = 0.81; (b) Satisfaction with Life Scale (SWLS) (α = 0.82) [25], a 5-item questionnaire to assess satisfaction with the respondent’s life as a whole, and c) SF-36 Health Survey [28], a standard measure for health-related quality of life, with 36 items and 9 dimensions (α = 0.86–0.87) concerning different aspects of mental and physical general health. Other secondary outcomes were selected by evaluating psychological constructs that have been previously linked to microbiome dynamics and that are negatively correlated to well-being, such as anxiety and emotional regulation. Anxiety was measured using The State-Trait Anxiety Inventory (STAI) [25]. This scale consists of 20 items (α = 0.92) that measure the current state of anxiety as a transient emotional state, characterized by subjective and perceived feelings of tension [29]. Our last secondary outcome was emotional regulation measured by Difficulties in Emotion Regulation Scale (DERS) [30], which assesses the subjective experience associated with emotional regulation in participants. This is a 25-item questionnaire (α = 0.92) with five dimensions [30].

Two exploratory outcomes were also included in order to investigate some aspects relating to top-down modulation of the MGBA, such as interoception and mindfulness. Alterations in interoceptive awareness have been proposed as an important feature of several mental health issues [31], several of which are also related to MGBA processes [32]. Interoception was assessed using the Multidimensional Assessment of Interoceptive Awareness (MAIA) questionnaire [33]. The scale has 32 items (α = 0.9) to measure multiple dimensions of interoception by self-report, and it has 8 dimensions. Mindfulness was measured using the Five Facet Mindfulness Questionnaire (FFMQ) [34], a questionnaire which consists of 39 items and 5 dimensions (α = 0.79).

Finally, the custom-made lifestyle survey was designed in order to study some aspects of participants’ lifestyles. This four-point Likert scale includes seven domains of lifestyle: diet, physical activity, sleep behavior, nature exposure, social contact, social media use, and substance abuse. To reduce the dimensionality of the variables, and considering that it has been observed that health-related behavioral patterns are generally associated between them across domains (diet, physical exercise, sleep, substance use, etc.) [35], we grouped them into two main patterns of behaviors: healthy and risky behaviors. We also grouped some items in a third category named undetermined behaviors. In the healthy behaviors pattern, a higher score represents health benefits. In the risk behaviors pattern, on the contrary, a higher score indicates detrimental effects of the behaviors. The “undetermined category” points to those categories that don’t fit into “higher is best” or “higher is worst” parameters according to the selected references. 

Healthy behavior patterns include the diet domain (whole grains, fruits, vegetables, omega 3, fermented foods), physical activity, nature exposure, and social contact. The dietary components of this category correspond to international recommendations [36,37,38,39,40] and are also coherent with the well-studied Mediterranean diet and Mediterranean-like evidence-based healthy dietary patterns [38]. Risk behavior patterns include diet categories (red meat, milk, snacks, refined flour, and sugar), social media use, and substance abuse (alcohol and smoking). The foods included in this category are restricted or limited by the previously referenced international scientific consensus. Similarly, social media have been proven as a risk factor for mental well-being [41,42], while alcohol and smoking behaviors are known as health-detrimental behaviors. Undetermined behaviors include diet categories (oils, dairies, and eggs) and sleep behavior. (See Appendix A for further details).

### 2.5. Statistical Analysis and Pre-Processing

Data from the full analysis set, including subjects randomized and exposed to products within the intervention arms, were analyzed according to the intention-to-treat principle. Estimates are given as the mean ± standard error/standard deviation, (SE/SD). A *p*-value < 0.05 was considered significant. Results are reported in accordance with 2010 CONSORT guidelines [43].

Data pre-processing and analysis, including correlations, principal component analysis (PCA), and mixed-effects models were performed on R Studio [44]. For overall results, Repeated Measures ANOVA were performed in Jamovi [45]. Plots of the results were made using the ggplot2 package [46]. Principal component analysis of numerical data was made with the “princomp” R function, while principal component analysis of categorical data was made with the Gifi package [47]. Bray–Curtis distance and principal coordinate analysis were performed using the “Vegan” R package [48]. Mixed-effects models were performed using the lme4 R package [49], while degrees of freedom and p-values were obtained using the lmerTest package [50], using the Kenward–Roger approximation in order to minimize the possibility of a type I error. Adjusted p values were obtained with the “stats” R package.

For bivariate analysis, such as correlation, we removed data points with z-scores that were greater than 3 or smaller than −3 standard deviations. For multivariate analysis, such as linear-mixed effect models, we removed multivariate outliers with a probability of occurring equally to or less than 1%, according to the Mahalanobis distance of that data point to the group centroid. All variables were mean centered before the correlation and linear-mixed effect analysis.

To understand the relationship between lifestyle habits and psychological variables we reduced the dimensionality of the lifestyle variables using PCA. We grouped lifestyle variables based on available evidence, by whether they belonged to healthy, risky, or uncertain behaviors and ran a separate PCA for each of the three groups of variables. The component scores of the set of variables comprising healthy behaviors will be called “Health Behavior” (HB), the one from the set of variables comprising risk behaviors will be called “Risk Behavior” (RB) and the one from the set of variables comprising uncertain behaviors will be called “Uncertain behavior” (UB). Each of these three variables is a value that summarizes the lifestyle habits of each subject, according to the three categories of lifestyle habits. Details about the categorization and statistical procedures can be found in Appendix A.

## 3. Results

### 3.1. Participants, Protocol Adherence and Demographics

As illustrated in Figure 2, a total of 827 voluntary subjects were assessed for eligibility, resulting in 143 meeting the inclusion criteria. 71 participants were randomly allocated to probiotic treatment and 72 to the placebo control group by a third-party researcher. In the probiotic group, 3 participants did not receive the treatment due to the following reasons: started an antibiotic treatment (*n* = 1), started a probiotic treatment (*n* = 1), and diagnosed COVID-19 (*n* = 1). Thus 68 participants received probiotic intervention and responded to the set of scales. In the placebo control group, 3 participants did not receive the treatment because of antibiotic treatment (*n* = 2) and a personal trip (*n* = 1). One participant discontinued the treatment due to adverse effects and another one due to personal issues, so 67 participants completed the study procedures. The mean rate of probiotic/placebo intake was 95.12% with a standard deviation of 5.53 in the overall sample (94.94 ± 5.88 in the probiotic group, and 95.31% ± 5.20 in the placebo group).

Participants’ ages ranged between 20–66 years (Overall: 33 ± 7.7 years; Placebo: 33.33 ± 8.83 years; Probiotic: 32.68 ± 6.43 years). Men were underrepresented with 20% of the total sample (Overall: 27 (20%); Placebo: 13 (19.4%); Probiotic: 14 (20.6%)). Finally, the mean BMI was 22.85 ± 3.56 (Placebo: 23.29 ± 3.5; Probiotic: 22.4 ± 3.6). Demographic statistics are shown in Table 1. All participants were considered in the analysis according to Intention-to-treat criteria. Additionally, in order to confirm that randomization was achieved successfully, we implemented a statistical procedure available in Appendix A. In short, we used two PCA approaches considering the scores of each participant on the 10 scales and observed no significant differences between both probiotic and placebo groups at baseline.

### 3.2. Effects of Probiotic Intervention on Well-Being

See Table 2 for main and interaction effects for the primary outcome (RYFF), secondary outcomes (SWLS, PANAS, STAI, SF-36 and DERS) and exploratory outcomes (MAIA and FFMQ). For each scale, Repeated Measures ANOVA was performed with all participants who completed the three time points according to the original group assignment regardless of the level of adherence. Although this is a standard procedure, it has the limitation of not including all subjects. In order to gain a better picture, alongside following the Intention-To-Treat (ITT) principle, we performed a mixed effect analysis which allowed us to include all subjects independently of missing data and controlling for more factors. Outcomes of the ITT analyses included all participants (*n* = 135) according to their original group assignment, regardless of level of adherence to the treatment or missing data. We conducted an analysis to evaluate the effectiveness of the probiotic treatment. No group differences were observed at baseline time for any scale (See Appendix A).

#### 3.2.1. Primary Outcomes

RYFF showed significant change over time (F(2,252) = 115.87, *p* < 0.001), but not between treatments nor interaction time*treatment (Table 2). Post-hoc pairwise correction analysis showed a significant increase in RYFF scores between baseline and follow-up times (M diff = −0.499, t(126) = −11.57, *p* < 0.01), and between post-treatment and follow-up (M diff = −0.444, t(126) = −13.26, *p* < 0.01), while no effect was observed between baseline and post-treatment times. Overall, our results demonstrate that while there were no significant effects derived from probiotic nor placebo treatments over RYFF scores, there was a significant score increase for both groups over time (Table 2).

#### 3.2.2. Secondary Outcomes

Five secondary scales for measuring aspects related to well-being were considered in order to evaluate the participant’s overall psychological state, using SWLS, the positive and negative subscales of PANAS, DERS, and the mental and physical components of SF-36. Overall, secondary outcomes revealed that during the experiment, participants exhibited changes in their psychological state that did not depend on the treatment group they were part of. All changes across time reflect an overall tendency to improve well-being and decrease anxiety and difficulties in emotional regulation, with the exception of the physical symptoms component of SF36, which tended to decrease in the post-treatment measures from baseline levels.

#### 3.2.3. Exploratory Outcomes

We explored the extent to which probiotic intervention may impact body awareness through two exploratory outcomes pertaining to interoceptive awareness and mindfulness facets. Similarly to primary and secondary outcomes, exploratory outcome analysis did not show any significant effects of treatment, suggesting that the probiotic intake did not impact body awareness when all participants of each group were analyzed together.

### 3.3. Lifestyle Behavior Impacts Psychological Measures

The lifestyle survey revealed participants’ lifestyle patterns (detailed account of lifestyle habits can be found in Appendix A). To understand the relationship between lifestyle habits and psychological variables, the PCA component scores were correlated with the psychological scales. Since there is evidence that lifestyle variables are strongly correlated with psychological variables [51], in order to correct for multiple comparisons we are going to consider a correlation significant if its *p*-value is below or equal to 0.002. This correction criterion is equal to a Bonferroni correction for 23 comparisons, as we are interested in the relationship between healthy and risk behaviors with the other 10 psychological variables.

As can be seen in Figure 3, healthy behaviors were significantly correlated with the RYFF scales (r(133) = 0.34, *p* < 0.001), SWLS (r(133) = 0.28, *p* < 0.001), STAI (r(133) = −0.43, *p* < 0.001), PANAS POS (r(133) = 0.41, *p* < 0.001), PANAS NEG (r(133) = −0.27, *p* = 0.002), MAIA (r(133) = 0.4, *p* < 0.001), FFMQ (r(133) = 0.39, *p* < 0.001), SF36 Physical (r(130) = 0.44, *p* < 0.001) and SF36 Mental (r(133) = 0.46, *p* < 0.001). Risk behavior was significantly correlated with RYFF (r(133) = −0.27, *p* = 0.002), STAI (r(133) = 0.3, *p* < 0.001), MAIA (r(133) = −0.3, *p* < 0.001), FFMQ (r(133) = −0.34, *p* < 0.001) and SF-36 Mental (r(133) = −0.29, *p* < 0.001). Uncertain behavior doesn’t correlate with any psychological variable, so that variable will not be analyzed further. The complete correlation table can be found in Appendix A.

Lifestyle variables were measured at three time points. To explore how participants’ lifestyles changed over time, we performed a mixed-effects model with health and risk behaviors as dependent variables. For each model, time, treatment, and their interaction were fixed effects and subjects were modeled as random effects. We also included the variables of sex and gender as fixed effects in order to control for them. For ease of interpretation, we used the ANOVA function over the mixed-effect model. Health behavior did not show any significant effect for treatment (F(1,131.04) = 0.01, *p* = 0.93), time (F(2,255.39) = 0.004, *p* = 1) nor their interaction (F(2,255.38) = 0.05, *p* = 0.95). Similarly, risk behavior did not show any significant effect for treatment (F(1,131.04) = 0.01, *p* = 0.93), time (F(2,255.27) = 0.0, *p* = 1) nor their interaction (F(2,255.26) = 1.34, *p* = 0.26). These results indicate that subjects did not significantly change their lifestyle habits across the measures. The results of the complete models can be found in Appendix A.

### 3.4. Lifestyle Behaviors Interact with Probiotic Effects

As the HB pattern showed significant associations with mental health, further analyses were conducted on this aspect. We had previously hypothesized that the effects of the probiotic could be associated with lifestyle patterns of the participants. To explore this association, we generated a separate mixed-effect model for each one of the psychological variables. Each model included as a fixed effect the subject score of healthy behaviors (HB), time (baseline, post, and follow-up), treatment group (probiotic/placebo) and the three-way and two-way interactions between these variables. Subjects were included as a random effect. In each model, we also controlled for age and gender.

Our parameters of interest were the interaction between treatment groups, lifestyle habits, and time. This three-way interaction reflects how the relationship between variables changes across time. Since there is not a clear view of the effect of probiotics on psychological outcomes [6,52], we corrected the *p*-values across the 10 models with the False Discovery Rate (FDR) correction in order to account for multiple comparisons while avoiding small effects.

We first analyzed whether health behavior interacted with the effects of probiotic treatment in predicting change between the baseline and the post-treatment time. The estimate of this three-way interaction is shown in Table 3. After FDR correction, we observed a significant interaction between adopting a healthy behavior and the protective effects of the probiotic intake. Participants with higher scores in HB had better outcomes only after the probiotic intake, in which they exhibited less difficulty in emotional regulation, measured as decreased scores in the DERS (b = −2.75, 95% CI [−4.95, −0.56], t(250.83) = −2.47, .p adjusted = 0.04, Figure 4a), and decreased anxiety as measured by the STAI scale (b = −2.68, 95% CI [−4.64, −0.73], t(257.23) = −2.70, .p adjusted = 0.04, Figure 4b). This result, although obtained from a sample from the healthy general population, suggests that the adoption of healthy habits could potentiate the beneficial effect of the psychobiotics in populations with specific conditions or with higher stress levels. Additionally, healthy behavior was also related to a probiotic-dependent increase in mindfulness attitude, as participants who ranked high in HB showed increased scores on the FFMQ scale (b = 3.93, 95% CI [−0.70, 7.17], t(252.51) = 2.39, .p adjusted = 0.02, Figure 4c), suggesting that healthy habits potentiate the probiotic’s beneficial effect on mindfulness. Finally, it is worth mentioning that we also observed a significant correlation between HB and the effects of the probiotic over the PANAS_NEG (b = −2.15, 95% CI [−3.70, −0.60], t(248.22) = −2.73, .p adjusted = 0.01, Appendix A). However, as mentioned above, given the suspicion that this instrument was not implemented properly, we will not delve into this interpretation.

We later analyzed whether health behavior interacted with the probiotic in predicting change between the baseline and the follow-up time. The estimate of this three-way interaction can be seen in Appendix A. No difference in scores between the baseline and follow-up time were found for any of the psychological variables. The complete output of each of the 10 mixed effect models and the plots of the other 6 scales can be found in Appendix A, respectively. Running the same analysis with risk behavior instead of health behavior as a fixed effect also did not result in any significant effect. The output of these analyses can be found on Appendix A.

Collectively, these results suggest that lifestyle habits should be considered when assessing the positive impact of a probiotic formulation on well-being and mental process. Alternatively, the adoption of healthier lifestyle habits could be included by design alongside probiotic intake. Importantly, as observed in this population, some beneficial effects might be masked if considering all participants together, especially when studying general population samples considered as otherwise healthy.

## 4. Discussion

In this randomized, double-blind placebo-controlled trial with healthy adults, we found no significant effect of probiotics on a set of psychological measures of well-being, anxiety, emotional regulation, interoception, and mindfulness in a general population sample qualified as otherwise healthy. However, further analyses revealed there was a relevant role of lifestyle behaviors concerning both overall mental health and probiotic effectiveness. Healthy Behaviors (HB) was positively correlated with well-being questionnaires while negatively correlated with anxiety and negative affect. Moreover, the interaction between high scores in HB and probiotic intake was the single significant predictor of positive effects on anxiety, difficulties in emotion regulation, and mindfulness in the post-treatment outcomes when controlling for sex and age.

Our study contributed to expanding the evidence on the gap concerning the effect of probiotics on mental health in healthy volunteers. Furthermore, our analyses identified variables influencing individual variation in response to treatment. Several meta-analyses point that there is still a lack of evidence for reaching strong conclusions about the efficacy of psychobiotics in healthy volunteers [53,54,55]. For example, meta-analytic evidence of RCTs looking at the benefit of probiotics on anxiety symptoms is still inconclusive and contradictory [15,55], thus it is suggested more rigorous research needs to be done considering the role of potential confounders. Interestingly, Liu et al. [15] aimed to analyze different variables that could influence probiotics’ effect on anxiety (health status of subjects, existence of gastrointestinal symptoms, intervention duration, strains of flora, risk of bias assessment), failing to identify a significant one. So, there is a recognized need to explore the hidden factors behind contradictory trial results and meta-analytic conclusions. Our research addresses this gap with a novel and evidence-based approach.

In our research, higher scores on our HB construct were significantly correlated with higher levels of well-being and lower levels of anxiety across our set of measures. Our study confirms previous findings about strong correlations between lifestyle factors and mental health [51,56], supporting the role of daily behaviors as important salutogenic factors. For example, Pano et al. [51] analyzed a Spanish cohort of 15,674 subjects, concluding that a healthy diet and lifestyle (physical activity and sleeping hours) have a quantifiable, direct association with health-related quality of life, a variable commonly related to well-being. Furthermore, they also concluded that poor dietary quality and below the recommended daily physical activity were negatively correlated with the overall perceived health of an individual. Although our HB scores are not directly comparable to those of Pano et al., our findings are coherent with their conclusions. It is worth mentioning that in our study, RB holds only small correlations with mental health indicators, which contrasts with previous research linking smoking and alcohol behavior to a worsening in several mental health indicators [57]. This might be related to the under-representation of regular smoking behavior and problematic alcohol consumption in our sample (See Appendix A).

Although research on healthy lifestyles commonly addresses diet, exercise and substance misuse, to study microbiome dynamics, environmental exposures should be considered [13]. For this reason, our HB construct also included how regularly people are exposed to natural environments. Green spaces have been related to better physical and mental health [58]. Some of the mechanisms proposed to underlie this association might be a decrease in cortisol levels, a conscious counteraction of the negative effects of stressful events, and an increasing general psychological well-being. Usually, these effects are attributed to the visual experience of green spaces, its synergistic relationship with physical exercise and its association with enhanced social cohesion [58]. In addition, growing research has proposed the microbiome as a potential mediator of the health benefits of visiting green spaces [59]. For example, Brame et al. [60] examined the potential of butyrate-producing bacteria colonizing the human gut, both through exposure to air microbiome (airbiome) as well as cutaneous exposure to soil. In mice, air exposure to soil dust with high microbial diversity induces anxiolytic properties in mice [61]. In humans, daily topical application of a biodiverse mixture, transiently increased alpha diversity of fecal microbiota [62]. This evidence suggests that exposure to natural environments could interact with human health and well-being through changes in gut microbiota.

Overall, our correlational results strongly support the growing body of evidence pointing to the relevance of diet and other lifestyle habits on mental health treatments, in line with the approaches of nutritional and lifestyle psychiatry [63,64]. The biological mechanisms by which these bottom-up influences can be understood comprises inflammation, oxidative stress, microbiota modifications, and epigenetic pathways, among others [65].

When considering the whole population, we observed no significant difference between groups on well-being or emotional regulation and anxiety. This observation is in line with recent meta-analyses conclusions identifying inconsistent effects of probiotics regarding pooled meta-analytic outcomes and denoting the need for further research to determine appropriate dosage, strains, and intervention duration between healthy and clinical samples [6,52,66]. Other meta-analyses have concluded low to moderate effects of probiotics on diminishing clinical and subclinical symptoms of depression, anxiety, and perceived stress [55,67], also highlighting the need to further understand the underlying mechanisms of the individual variation.

It is possible that insufficient sample size or intervention duration have played a role in the apparent lack of effect when considering the whole population. It is also possible, as our exploratory analyses suggest, that our broad notion of “healthy volunteers” and our inclusion and exclusion criteria did not define a sufficiently specific sample. We discuss this in the following section. It is also relevant to note that this randomized clinical trial was done in the early stage of the COVID-19 pandemic in Chile. This period has been associated with high levels of psychological distress among Chileans, especially in women [68], which constitute 80% of our sample. It could be possible that these conditions may have had an influence on the overall effects of the treatment, also given that previous evidence on the same product have suggested that stress levels may influence its efficiency [21]. Moreover, these considerations help to make sense of the overall decrease in self-perceived physical health as measured by the physical component of SF36, given a social context that was increasingly concerned about the presence of physical symptomatology.

The same probiotic formulation we tested has previously shown potential to exert anti-stress and anti-anxiety effects on healthy subjects [14] and secondary analyses suggested that probiotic efficiency may be modulated by individual’s stress levels, concluding that this product could benefit well-being especially in those subjects with lower levels of stress [21]. We hypothesized that certain lifestyle-dependent physiological conditions might influence psychobiotics effectiveness. We analyzed whether the level of healthy behaviors (HB) interacted with the probiotic in predicting change between baseline and the post-treatment time. Our results showed that the high prevalence of HB predicted the efficacy of probiotics on diminishing anxiety, difficulties in emotional regulation, and negative affect, together with increasing mindfulness. In other words, the beneficial effect of the test product was dependent on the level of HB.

A way to enhance probiotic effectiveness is the combination of probiotics and prebiotic compounds, under the concept of ‘synbiotic’ [69]. Synergistic synbiotics refer to the kind of action our findings suggest, where a combined effect is better than the effects of each component separately. For example, co-supplementation of probiotics and prebiotics have shown positive effects on depressive symptoms and inflammation markers in patients with coronary artery diseases [70]. In the case of this study, the probiotic component of this relationship corresponded to the test product and the prebiotic component could plausibly come from dietary sources. Diet patterns are in fact a well-established modulator of the gut microbiota [71,72]. From this dimension, one of the clearer ways of understanding the mediating role of healthy behaviors on psychobiotic effect is related to prebiotic-richer diets [73]. Prebiotics fibers are the main fermentable substrates from which psychobiotic microorganisms thrive, such as Lactobacillus and Bifidobacterium strains. In healthier diets containing sufficient fermentable fibers, the tested probiotic species could better establish themselves and thrive. It has been proposed that microbiota capacity to adequately ferment prebiotic fiber could benefit from the introduction of fiber-consuming microbes [74], e.g., probiotics consumption. Probiotics can increase microbiota diversity and therefore maximize the production of neuroactive compounds derived from fiber fermentation, such as short-chain fatty acids (SCFA), e.g., butyrate and acetate, therefore allowing the established dietary patterns to exert significant beneficial effect on brain function [11,75], as has been shown in previous interventional research [76]. Greater microbiota diversity is associated with lower levels of systemic inflammation [74,77] and improvements in conditions such as depression and anxiety [65], and it may be a plausible mechanism to understand the lifestyle-dependent effects of psychobiotics in our study.

Our findings are also in agreement with previous interventional research showing a high level of inter-individual variability in the response to probiotics [10,78]. Previously, such variability has been analyzed via specific symptomatology revealing that a whole sample effect is in a certain way masking more population-specific benefits. For example, Santocchi et al. [78] found that beneficial effects of a probiotic formulation on the severity of Austistic Spectrum Disorder symptomatology in children were specific to the subgroup with gastrointestinal symptoms. Similarly, Wauters et al. [10] found significant decreases in subjective anxiety after 4-weeks of supplementation with *Lactobacillus rhamnosus* CNC I-3690 in students exposed to an academic stressor, but the protective effect of the probiotic over subjective stress was restricted only to a subpopulation with a stress-induced cortisol response higher than the average. As we previously hypothesized, it is plausible that lifestyle factors such as diet, physical activity or exposure to natural environments exert some type of regulation over probiotic effectiveness, given the well-documented effects these habits have on the microbiome [18,59,79,80]. In fact, lifestyle behaviors have been referred to as potential confounders of probiotic effects on cognitive functions and the scientific literature have consistently highlighted the need for lifestyle behaviors to be considered as control variables in microbiome research [67,79,81,82,83,84].

Lifestyle habits such as diet, exercise, and smoking are established determinants of health [51,57]. Although there is a clear focus on the high prevalence of non-communicable chronic diseases within the research on lifestyle behaviors (heart disease, type 2 diabetes, and cancer), evidence supports the urge to include common mental health problems such as depression and anxiety in this rank [85]. Moreover, lifestyle changes are being proposed as prevention and treatment strategies for mental health in general [85,86]. In the last years, some RCTs have shown important mental health benefits from nutritional interventions [87,88] and physical exercise interventions [64] demonstrating that brain function can be modified through lifestyle changes. All these effects have been associated with potential changes in gut microbiota.

All in all, there are reasons to extend lifestyle characterization of trial participants to other mental health interventions, such as psychopharmacology, psychotherapy, and other evidence-based interventions, for example, mindfulness-based stress reduction programs. Previously, controlling for lifestyle patterns in mental health research has been proposed as a way to face the problem of heterogeneity of outcomes in clinical trials concerning mental health [89]. However, to the best of our knowledge, there is little research considering these factors to this date. In response to this challenge, our analyses embrace an embodied, extended, and embedded account of cognitive processes, in which the mutual influence between lifestyle patterns and microbiome dynamics are not understood as peripheral factors, but as constituent parts of mental processes [2].

Interestingly in this line, the positive effects on mindfulness traits as measured by FFMQ represent a novel finding. The literature linking mindfulness and the microbiota-gut-brain axis is scarce. Moreover, the focus of existing research relates to top-down processes, i.e., the potential benefit of mindfulness practices on Irritable Bowel Diseases [90]. In contrast, our findings suggest the possible bottom-up effects, i.e., that physiological changes (in this case, microbiota changes) could induce transient increases in mindfulness facets. Although these are preliminary findings, it is known that MGBA dynamics are bi-directional and there are existing physiological mechanisms that could explain this kind of interaction via microbiota-mediated interoceptive signaling.

Finally, in this study, we hypothesized that the impact of microbiome modifications on psychological functioning may be modulated by people’s lifestyle. We did so grounded in previous theoretical proposals that approach the complex relationships between the mind, physiological conditions and environmental conditions as a way to accomplish a more integrative view of cognition and therefore of mental health. Our findings show that indeed the effects of the tested psychobiotic formula were potentiated by healthy behaviors in each individual, which are associated with specific microbiome and immunological signatures. Moreover, without considering this behavioral condition of the participant´s physiology, there appeared to be no significant effects of the treatment. In addition to that, we identify a need for further assessing macro-societal factors influencing health perception and general well-being of the participants (e.g., COVID). It is important to develop methodological designs and viable assessment tools to be able to account for these constituent factors of health and cognition.

There are some limitations that should be highlighted in order to improve future research and motivate scientific transparency. First, our sample was predominantly women, which had been identified as a predictor of higher psychological distress in this time period [68]. Our sample was also ‘biased’ to a Mediterranean-like diet and lifestyle, which may affect the generalizability of our conclusion. Second, regarding our analytical approach, we are currently underpowered in the scope of our mixed-effect model; in order to appropriately detect a three-way interaction, the appropriate sample size is fourfold of what is required for a two-way interaction in a repeated measures design [91]. Therefore, studies with bigger sample sizes are needed to confirm our results. Third, future works might consider more exhaustive and extensive lifestyle assessments; however, this could significantly increase the response-time duration, which could also be problematic. Finally, we were not able to include microbiome data in the current manuscript, which would have increased the explanatory/mechanistic value of this report. Moreover, including the microbiome data it has would have been useful for discussing the ‘lack’ of the main effect. In particular, the effectiveness of the intervention in inducing the desired microbiota changes could be questioned. However, the trial involved the probiotic concentration that has been previously used and suggested by the literature [21,92]. The lack of microbiome data does not obscure the clinical relevance the trial provides, which highlights lifestyle behaviors are relevant confounders that should be included in probiotic interventions (and also considered in other type of interventions)

Several strengths can be also identified. First, this study has a strong study design in terms of randomization and blindness which allowed us to reach valid conclusions. Second, this is the first study –to the best of our knowledge– that considers a possible interaction between probiotics and healthy lifestyle behaviors, opening a huge window to future research within the area and outside of it. Additionally, there was a high adherence and low drop out in both groups (See Figure 1). Finally, the diversity of scales used to assess psychological well-being embraces the multidimensionality of the construct from a broader perspective.

The findings of this study support the notion that lifestyle behaviors are important factors that may modulate psychobiotics effectiveness on anxiety and emotional regulation. Therefore, it is crucial to control these behavioral variables in future studies, both as a way to identify the specific populations that can benefit the most from psychobiotic treatments and as a way to advance existing knowledge about the role of MGBA dynamics upon mental health and disease. We also identify a need to take more into account biological and physiological parameters such as microbiota composition index and lifestyle behavioral patterns in other subfields of mental health research and practice.

There are several implications of our results aside from the already mentioned need to better control for lifestyle in clinical trials. As we have discussed, lifestyle patterns are relevant both to psychological outcomes and to psychobiotic treatment response. From the clinical practice perspective, our results support the emphasis of recent psychiatry guidelines [93,94] and the nutritional and lifestyle psychiatry approach [64,86] to assess and prescribe lifestyle behaviors as a primary strategy, under both preventative and treatment approaches. The physiological processes triggered by lifestyle patterns could plausibly impact not only probiotic supplementation response, but also established psychopharmacological interventions [89]. Therefore, efforts must be made to better establish this understanding. Moreover, focusing more efforts in promoting behavioral change towards healthy lifestyles can benefit well-being and a broad range of mental health conditions, but also an even greater range of non-communicable chronic diseases such as some kinds of cancer, diabetes type-II, and coronary heart diseases [85]. In this sense, lifestyle medicine represents a modality of intervention to further integrate physical and mental health care. Our results highlight the importance of a personalized approach at the moment of designing probiotic interventions, supporting the idea of unique intervention based on particular subjects’ microbiome and habits, namely personalized medicine [95]. Concerning the implications of the specific effect on anxiety and emotional regulation symptoms in our sample, it should be noted that the interpretations of our results on healthy volunteers are possibly not generalizable to clinical population, in part for potential differences in intervention response and in part for the different function that the specific symptomatology may have when considering the broader psychological functioning and psychosocial dimensions of experience of each individual. However, more research is needed to clarify this issue.

Overall, we found preliminary data that highlight how psychological functioning is dependent upon participant’s lifestyle behavioral patterns. Moreover, our exploratory analyses suggest a bottom-up influence of psychobiotics on anxiety, emotional regulation, and mindfulness traits and interestingly, this effect was dependent on the level of adherence to healthy behaviors. Our results demonstrate the need for controlling for lifestyle variables as a standard practice in human microbiome research and mental health research in general. These findings are also in accordance with an embodied account of psychological states that could support future interdisciplinary developments in mental health research and clinical practice.

## Figures and Tables

**Figure 1 nutrients-15-01706-f001:**
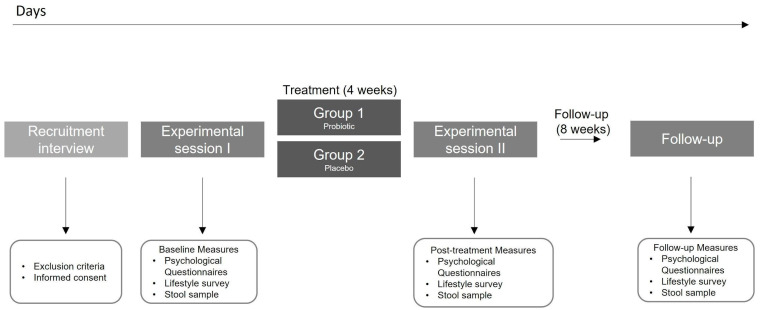
Study protocol.

**Figure 2 nutrients-15-01706-f002:**
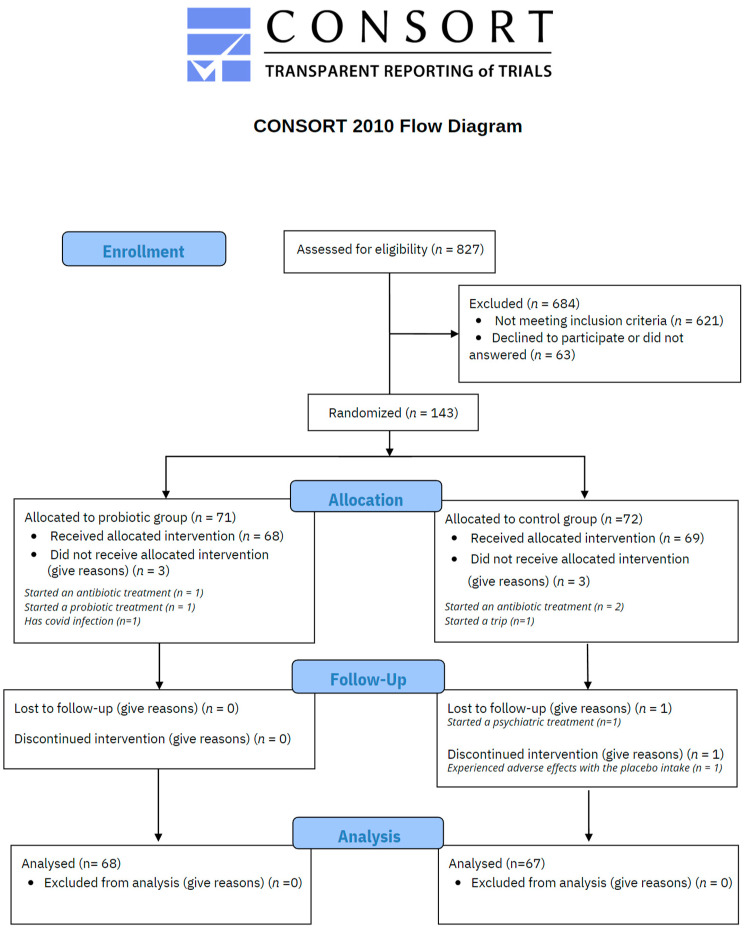
CONSORT flow chart.

**Figure 3 nutrients-15-01706-f003:**
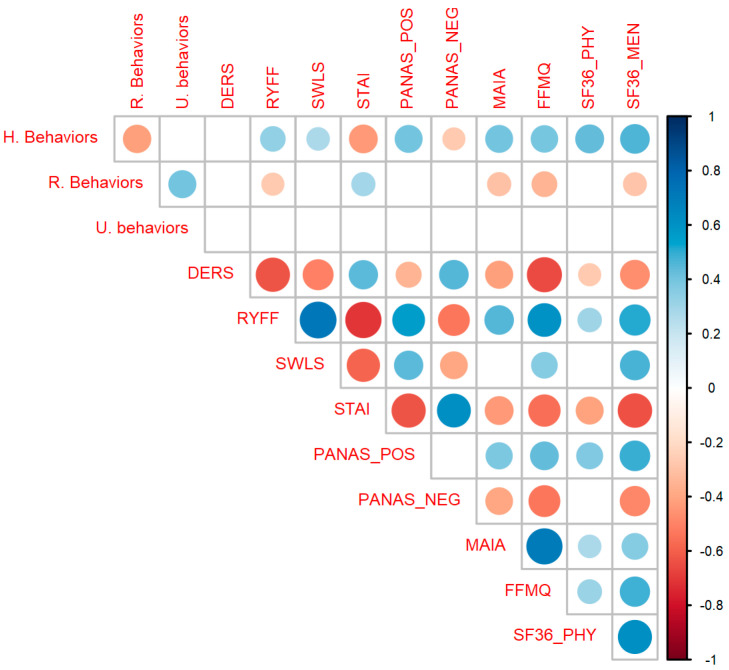
Correlation matrix between all variables. Healthy Behaviors (HB) significantly correlates with almost all psychological scales as expected (positively with well-being and interoceptive variables and negatively with anxiety and negative affect). Circles represent significant correlations after correcting for multiple comparisons. Colors blue and red indicate positive and negative values of r, respectively. The size of the circles and the intensity of each color represents the magnitude of the correlation. Abbreviations: STAI, State Trait Anxiety Inventory; RYFF, Ryff Well-being Scale; DERS, Difficulties in Emotional Regulation Scale; SWLS, Satisfaction with Life Scale; PANAS_POS, Positive and Negative Affect Scale, subscale Positive; PANAS_NEG, Positive and Negative Affect Scale, subscale Positive; SF36_PHY, Short Form Health Survey, subscale Physical; SF36_MEN, Short Form Health Survey, subscale Mental; FFMQ, Five Facet Mindfulness Questionnaire; MAIA, Multidimensional Assessment of Interoceptive Awareness Scale.

**Figure 4 nutrients-15-01706-f004:**
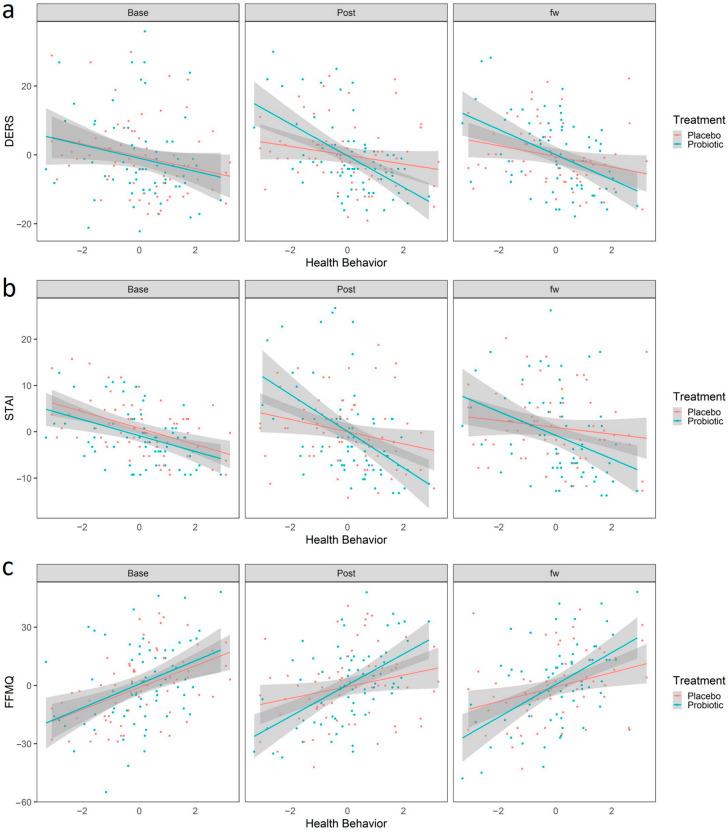
Linear Regression predicted scores using the linear regression model “Scales~Sex + Age + HB + Treatment+ HB × treatment + HB × treatment × time”. Red and blue colors represent placebo and probiotic groups, respectively. Points represent the values for DERS (**a**), STAI (**b**) and FFMQ (**c**). The shaded region represents the standard error. Abbreviations: STAI, State Trait Anxiety Inventory; DERS, Difficulties in Emotional Regulation Scale; FFMQ, Five Facet Mindfulness Questionnaire; HB, Healthy Behaviors.

**Table 1 nutrients-15-01706-t001:** Baseline measurements.

Group	Total *n* = 135	Placebo *n* = 67	Probiotic *n* = 68	*p*
Demographics				
Age (mean std)	33 (7.7)	33.33 (8.83)	32.68 (6.43)	0.89
Male (total %)	27 (20)	13 (19.4)	14 (20.6)	0.87 *
BMI (mean std)	22.85 (3.56)	23.29 (3.5)	22.4 (3.6)	0.72

Abbreviations: BMI, Body Mass Index; *p* Value is de result of Mann-Whitney U Test or Chi Squared Test (*).

**Table 2 nutrients-15-01706-t002:** Main and interaction effects for overall outcomes in overall analysis (Repeated Measures ANOVA).

Effect	Time		Treatment		Time × Treatment	
Outcome	F	*p*	F	*p*	F	*p*
Primary						
RYFF	115.87	<0.001	1.3	0.25	1.05	0.35
Secondary						
SWLS	7.76	<0.001	3.83	0.05	0.03	0.96
PANAS_POS	0.07	0.92	1.44	0.23	0.71	0.48
PANAS_NEG	3.17	0.04	3.55	0.06	3.53	0.03
STAI	19.11	<0.001	1.79	0.18	1.00	0.36
SF36_MEN	6.82	0.001	0.29	0.59	0.22	0.80
SF36_PHY	5.29	0.006	0.86	0.35	0.58	0.56
DERS	12.89	<0.001	0.12	0.72	0.08	0.91
Exploratory						
MAIA	7.27	<0.001	2.77	0.09	1.61	0.20
FFMQ	6.84	0.001	0.04	0.83	0.03	0.96

Abbreviations: RYFF, Ryff Well-being Scale; STAI, State Trait Anxiety Inventory; SWLS, Satisfaction with Life Scale; PANAS_Pos, Positive and Negative Affect Scale (Positive); PANAS_Neg, Positive and Negative Affect Scale (Negative); SF36_PHY, Short Form Health Survey (Physical); SF36_MEN, Short Form Health Survey (Mental); DERS, Difficulties in Emotional Regulation Scale; MAIA, Multidimensional Assessment of Interoceptive Awareness; FFMQ, Five Facet Mindfulness Questionnaire.

**Table 3 nutrients-15-01706-t003:** Mixed-effects model interaction estimates between Health Behavior, Probiotic intake and time for each scale.

Scale	beta Coeff.	Std. Error	CI	t Statistic	df	*p*	p Adjusted
DERS	−2.75	1.11	−4.95–−0.56	−2.47	250.87	0.01	0.04
RYFF	0.06	0.05	−0.04–0.15	1.17	255.02	0.25	0.28
SWLS	0.52	0.35	−0.17–1.22	1.48	249.64	0.14	0.23
STAI	−2.68	0.99	−4.64–−0.73	−2.70	257.23	0.01	0.04
PANAS_POS	−0.32	0.88	−2.05–1.40	−0.37	255.09	0.71	0.71
PANAS_NEG	−2.15	0.79	−3.70–−0.60	−2.73	248.22	0.01	0.04
MAIA	−0.08	0.06	−0.20–0.05	−1.23	254.65	0.22	0.28
FFMQ	3.93	1.64	0.70–7.17	2.39	252.51	0.02	0.04
SF36_PHY	1.69	1.00	−0.28–3.66	1.69	247.27	0.09	0.18
SF35_MEN	1.84	1.59	−1.29–4.96	1.16	253.09	0.25	0.28

Abbreviations: STAI, State Trait Anxiety Inventory; RYFF, Ryff Well-being Scale; DERS, Difficulties in Emotional Regulation Scale; SWLS, Satisfaction with Life Scale; PANAS_POS, Positive and Negative Affect Scale, subscale Positive; PANAS_NEG, Positive and Negative Affect Scale, subscale Positive; SF36_PHY, Short Form Health Survey, subscale Physical; SF36_MEN, Short Form Health Survey, subscale Mental; FFMQ, Five Facet Mindfulness Questionnaire; MAIA, Multidimensional Assessment of Interoceptive Awareness Scale.

## Data Availability

The data and scripts that support the findings of this study are available upon reasonable request, at https://osf.io/hsey2/ (accessed on 22 March 2023).

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
