# Peer review of "Psychobiotic Effects on Anxiety Are Modulated by Lifestyle Behaviors: A Randomized Placebo-Controlled Trial on Healthy Adults"

_nutrients, 2023, doi:10.3390/nu15071706_

Round 1

Reviewer 1 Report

This is a well-written and structured manuscript on a RCT providing a 4-week probiotic intervention to healthy people. The authors did not find an impact of the probiotic treatment compared to the placebo group on well-being, quality of life, and emotional regulation. However, they described effects of lifestyle on the well-being and interactions with the intervention.

In my opinion, the topic of the analysis is important and worth publishing due to described reasons by the author. All in all, I would like to congratulate the authors on this nice work.

The only adaption I would suggest is a more detailed description in the methods part on the lifestyle assessment. At the moment the authors only refer to the supplementary material, but they should present clear reasons with references for their chosen questions on the lifestyle (many questions for diet; only one for activity, etc). Furthermore, the description for categorizing healthy and risky behavior is missing.

I was wondering what the reasons for clustering in lifestyle categories were; in particular, as the authors discuss special parts (green spaces; diet; smoking; Alcohol) in the discussion part. Please clarify.

Author Response

Reviewer 1

We would like to express our gratitude to the reviewer for their valuable suggestions and comments, which have undoubtedly improved the quality of the current manuscript. In this response letter, we have addressed and clarified the reviewer's points. We divided the reviewer comments into 3 different points.

C.1

This is a well-written and structured manuscript on a RCT providing a 4-week probiotic intervention to healthy people. The authors did not find an impact of the probiotic treatment compared to the placebo group on well-being, quality of life, and emotional regulation. However, they described effects of lifestyle on the well-being and interactions with the intervention.

In my opinion, the topic of the analysis is important and worth publishing due to described reasons by the author. All in all, I would like to congratulate the authors on this nice work.

The only adaptation I would suggest is a more detailed description in the methods part on the lifestyle assessment. At the moment the authors only refer to the supplementary material, but they should present clear reasons with references for their chosen questions on the lifestyle (many questions for diet; only one for activity, etc). 

  1. Thank you for bringing this up. Initially, we only included this explanation in the Supplementary Material due to concerns about the length of the manuscript. However, we have now included it in the revised main text in response to your comment. Please refer to page 7 for further details.

In our lifestyle questionnaire, we include categories such as diet, physical activity, sleep behavior, nature exposure, social contact, social media use, and substance abuse. These categories are based on the framework of lifestyle medicine, which is an evidence-based approach to the prevention and treatment of chronic diseases (1-2). While the most researched lifestyle factor in relation to mental health and the gut microbiota is diet (3), we recognize that other lifestyle pillars also have significant influences on both microbiota dynamics (4-6) and mental health (2, 7).

We have focused more on diet in our questionnaire since it is the most researched lifestyle factor in this context. However, we recognize the importance of the other categories in understanding the complex relationships between lifestyle, mental health, and gut microbiota. 

  1. Sarris, Jerome, Adrienne O’Neil, Carolyn E. Coulson, Isaac Schweitzer, and Michael Berk. 2014. “Lifestyle Medicine for Depression.” BMC Psychiatry 14 (April): 107.
  2. Marx, Wolfgang, Sam H. Manger, Mark Blencowe, Greg Murray, Fiona Yan-Yee Ho, Sharon Lawn, James A. Blumenthal, et al. 2022. “Clinical Guidelines for the Use of Lifestyle-Based Mental Health Care in Major Depressive Disorder: World Federation of Societies for Biological Psychiatry (WFSBP) and Australasian Society of Lifestyle Medicine (ASLM) Taskforce.” The World Journal of Biological Psychiatry: The Official Journal of the World Federation of Societies of Biological Psychiatry, October, 1–54.
  3. Marx, Wolfgang, Melissa Lane, Meghan Hockey, Hajara Aslam, Michael Berk, Ken Walder, Alessandra Borsini, et al. 2021. “Diet and Depression: Exploring the Biological Mechanisms of Action.” Molecular Psychiatry 26 (1): 134–50.
  4. Mailing, Lucy J., Jacob M. Allen, Thomas W. Buford, Christopher J. Fields, and Jeffrey A. Woods. 2019. “Exercise and the Gut Microbiome: A Review of the Evidence, Potential Mechanisms, and Implications for Human Health.” Exercise and Sport Sciences Reviews 47 (2): 75–85.
  5. Han, Mengqi, Shiying Yuan, and Jiancheng Zhang. 2022. “The Interplay between Sleep and Gut Microbiota.” Brain Research Bulletin 180 (March): 131–46.
  6. Roslund, Marja I., Riikka Puhakka, Mira Grönroos, Noora Nurminen, Sami Oikarinen, Ahmad M. Gazali, Ondřej Cinek, et al. 2020. “Biodiversity Intervention Enhances Immune Regulation and Health-Associated Commensal Microbiota among Daycare Children.” Science Advances 6 (42). https://doi.org/10.1126/sciadv.aba2578.
  7. Firth, Joseph, Marco Solmi, Robyn E. Wootton, Davy Vancampfort, Felipe B. Schuch, Erin Hoare, Simon Gilbody, et al. 2020. “A Meta-Review of ‘Lifestyle Psychiatry’: The Role of Exercise, Smoking, Diet and Sleep in the Prevention and Treatment of Mental Disorders.” World Psychiatry: Official Journal of the World Psychiatric Association  19 (3): 360–80.

C.2

Furthermore, the description for categorizing healthy and risky behavior is missing.

  1. Thank you for bringing this to our attention. The description you mentioned was originally included in the Supplementary Material, but we have since moved it to the main text in the revised edition. You can find it on page 7 and 8.

To account for the various dimensions of lifestyle patterns, we classified the items into three categories based on international evidence-based guidelines (1-5): healthy behaviors, risk behaviors, and undetermined behaviors. Healthy behaviors consist of a Mediterranean-like diet (rich in whole grains, legumes, fruits, nuts, and vegetables), increased physical activity, social interaction, and exposure to nature. Risk behaviors include tobacco and alcohol consumption, social media use, and the consumption of sugary, fatty, and processed foods. Undetermined behaviors grouped items that didn´t match the other two categories.

Notably, after grouping the behaviors in this theoretical-based manner, we conducted a Principal Component Analysis (PCA)  to verify whether our data was consistent with this categorization. The results confirmed that it was (Please see figure S2).

  1. Marx, Wolfgang, Sam H. Manger, Mark Blencowe, Greg Murray, Fiona Yan-Yee Ho, Sharon Lawn, James A. Blumenthal, et al. 2022. “Clinical Guidelines for the Use of Lifestyle-Based Mental Health Care in Major Depressive Disorder: World Federation of Societies for Biological Psychiatry (WFSBP) and Australasian Society of Lifestyle Medicine (ASLM) Taskforce.” The World Journal of Biological Psychiatry: The Official Journal of the World Federation of Societies of Biological Psychiatry, October, 1–54.
  2. Cena, Hellas, and Philip C. Calder. 2020. “Defining a Healthy Diet: Evidence for The Role of Contemporary Dietary Patterns in Health and Disease.” Nutrients 12 (2). https://doi.org/10.3390/nu12020334.
  3. National Health and Medical Research Council (Australia), and Australia. Department of Health and Ageing. 2013. Eat for Health : Australian Dietary Guidelines ; Providing the Scientific Evidence for Healthier Australian Diets. [Canberra A.C.T.]: National Health and Medical Research Council.
  4. Ruano, Cristina, Patricia Henriquez, Miguel Ángel Martínez-González, Maira Bes-Rastrollo, Miguel Ruiz-Canela, and Almudena Sánchez-Villegas. 2013. “Empirically Derived Dietary Patterns and Health-Related Quality of Life in the SUN Project.” PloS One 8 (5): e61490.
  5. US Department of Agriculture and US Department of Health and Human Services. 2020. “Dietary Guidelines for Americans 2020–2025.” US Government Publishing Office. DietaryGuidelines.gov.

C.3

I was wondering what the reasons for clustering in lifestyle categories were; in particular, as the authors discuss special parts (green spaces; diet; smoking; Alcohol) in the discussion part. Please clarify.

Thank you for bringing up this issue. It was a topic of extensive discussion within our research group, and we arrived at the following reasoning for our decision to cluster lifestyle factors:

  • Previous research on probiotic trials and mental health has not accounted for lifestyle factors, making it challenging to tease apart their individual contributions.
  • Lifestyle variables influence individuals simultaneously and in a complex way, which poses a challenge to statistical modeling.
  • Health-related behaviors are often adopted in clusters, suggesting that grouping them together is reasonable (4)
  • To model the interaction between lifestyle variables while acknowledging their multidimensionality, we needed to reduce the dimensionality of the variables.

Regarding the inclusion of the paragraph highlighting research on the health effects of green spaces, our intention was to introduce this topic as a relevant area of study when considering microbiome dynamics and its relationship with health. This is a promising but nascent field of study (2-3). Based on your feedback, we have included a statement in the revised version of the manuscript (please see page 15) that clarifies this point and references a paper in which we propose a theoretical framework for understanding the relevance of environmental exposures to cognitive processes (4). We didn't discuss all lifestyle dimensions just for a matter of length. Each one of these dimensions is a whole field of research, we tried, however, to point at relevant literature where necessary.

  1. Noble, Natasha, Christine Paul, Heidi Turon, and Christopher Oldmeadow. 2015. “Which Modifiable Health Risk Behaviours Are Related? A Systematic Review of the Clustering of Smoking, Nutrition, Alcohol and Physical Activity (’SNAP') Health Risk Factors.” Preventive Medicine 81 (December): 16–41.
  2. Marx, Wolfgang, Sam H. Manger, Mark Blencowe, Greg Murray, Fiona Yan-Yee Ho, Sharon Lawn, James A. Blumenthal, et al. 2022. “Clinical Guidelines for the Use of Lifestyle-Based Mental Health Care in Major Depressive Disorder: World Federation of Societies for Biological Psychiatry (WFSBP) and Australasian Society of Lifestyle Medicine (ASLM) Taskforce.” The World Journal of Biological Psychiatry: The Official Journal of the World Federation of Societies of Biological Psychiatry, October, 1–54.
  3. Sudimac, Sonja, Vera Sale, and Simone Kühn. 2022. “How Nature Nurtures: Amygdala Activity Decreases as the Result of a One-Hour Walk in Nature.” Molecular Psychiatry 27 (11): 4446–52.
  4. Palacios-García, Ismael, Gwynne A. Mhuireach, Aitana Grasso-Cladera, John F. Cryan, and Francisco J. Parada. 2022. “The 4E Approach to the Human Microbiome: Nested Interactions between the Gut-Brain/body System within Natural and Built Environments.” BioEssays: News and Reviews in Molecular, Cellular and Developmental Biology 44 (6): e2100249.

Reviewer 2 Report

The authors describe a randomized probiotic trial in healthy individuals targeting well-being, quality of life, emotional regulation, anxiety, mindfulness and interoceptive awareness. The trial was sufficiently powered to detect any effects of the probiotic on primary as well as secondary outcome variables. However, it is unclear what specific improvements the authors expect in a healthy population as there is no indication that at baseline participants suffered from emotional dysregulation, anxiety etc. Rather than a healthy population the targeted population should have been one with concerns in any of the outcome variables. In a healthy population, changes in the outcome variables linked to the probiotic intervention, which weren’t observed here, would have been difficult to interpret. More or less anxiety at healthy levels likely still reflect healthy levels and a certain degree of anxiety is potentially beneficial rather than detrimental. While change in microbiota through probiotic supplementation is the obvious mechanism through which this intervention is supposed to work, the authors provide no data on microbiota dynamics in this manuscript. Even without efficacy in their primary or secondary outcome markers there is value in reporting the microbiota dynamics associated with the intervention. Maybe there was an effect on the outcomes in individuals in whom microbiota changes were observed during the trial (in either study arm). Without including data on microbiota dynamics there is little value to this current manuscript.  There are also some concerns with data presentation. In table 1 the authors provide p-values for the differences between placebo and probiotic group. This is nonsensical as we know the likelihood for the observed differences between the groups to be due to chance is actually 1, as subjects were randomized in this study (assigned into groups by chance). In the results section the authors state “the mean BMI was 22.58 ± 3.56 (Placebo: 23.29 ± 3.5; Probiotic: 22.4 ± 3.6)”. With equal sized groups it is unlikely that this data is accurate as the mean BMI should have been close to 22.85” Maybe just a typo, but it’s concerning that none of the authors detected such an obvious error.

Author Response

Reviewer 2

Before we start addressing the reviewer’s suggestions and comments, we would like to thank the reviewer. Addressing the following points has undoubtedly improved the quality of the current manuscript version. We divided the full paragraph in 4 main points that will be clarified in the following response letter.

C.1 

The authors describe a randomized probiotic trial in healthy individuals targeting well-being, quality of life, emotional regulation, anxiety, mindfulness and interoceptive awareness. The trial was sufficiently powered to detect any effects of the probiotic on primary as well as secondary outcome variables. However, it is unclear what specific improvements the authors expect in a healthy population as there is no indication that at baseline participants suffered from emotional dysregulation, anxiety etc. Rather than a healthy population the targeted population should have been one with concerns in any of the outcome variables. In a healthy population, changes in the outcome variables linked to the probiotic intervention, which weren’t observed here, would have been difficult to interpret. More or less anxiety at healthy levels likely still reflect healthy levels and a certain degree of anxiety is potentially beneficial rather than detrimental. 

  1. We really appreciate that this point came up. There was a very deep group discussion about this point during the trial design. We will explain our decision through three different arguments.
  • We do not understand mental health as a dichotomic process but rather as a continuum. Vikram Patel et al, at the Lancet Commission on global mental Health and sustainable development (1), recognizes that in between of ‘mental wellbeing’ and ‘mental syndrome’ there are plenty of shades. In this regard someone who is categorized as ‘healthy’ might significantly improve their wellbeing and mental health after some intervention or lifestyle modification. It is true that studying ‘non healthy population’ is extremely interesting, however, there are no reasons to think that a ‘healthy’ population will not benefit from a probiotic intervention. Indeed, using healthy population for assessing a probiotic effect is very common in different fields, and it is an interesting strategy to understand the clinical impact of them (2-3). Thus, our results can find already published literature to be compared and interpreted.

  • Currently, probiotics do not require a medical prescription, indeed they are mostly consumed as supplements with a preventive purpose. A huge survey from the International Probiotics Association in 2022 with around 8000 people from 8 different European countries revealed that probiotics consumers use probiotics for improving digestion, preventing disease, and because of their overall health benefits (4). Moreover, another survey from CHR Hansen revealed that 48% of the 15.000 people surveyed, consume probiotics daily (5). Finally, it is worth mentioning the probiotics market size was 48 billion dollars in 2019 and is projected to reach 94 billion by 2027 (6). Thus, as these data suggests, probiotics usage is mainly for preventive purposes and as complementary supplements, not really for facing particular health issues. For that reason, we believe that is relevant to increase evidence of what probiotics is doing on these healthy population. As probiotic research is mainly focused on clinical population, randomized placebo controlled trials on healthy people are missing, especially in the field of wellbeing and mental health.

  • Finally, as suggested in A, we do not assume that healthy people can not improve their physical and mental health. Similarly, we would not assume that little modifications on anxiety or other mental processes or constructs might be beneficial or detrimental since there are plenty of confounders that might be affecting this outcome. That point is precisely what we wanted to address in this report. 

  1. Patel, V., Saxena, S., Lund, C., Thornicroft, G., Baingana, F., Bolton, P., ... & UnÜtzer, J. (2018). The Lancet Commission on global mental health and sustainable development. The lancet, 392(10157), 1553-1598.
  2. Skonieczna-Żydecka, K., Kaźmierczak-Siedlecka, K., Kaczmarczyk, M., Śliwa-Dominiak, J., Maciejewska, D., Janda, K., ... & Łoniewski, I. (2020). The effect of probiotics and synbiotics on risk factors associated with cardiometabolic diseases in healthy people—a systematic review and meta-analysis with meta-regression of randomized controlled trials. Journal of clinical medicine, 9(6), 1788.
  3. Messaoudi, M., Violle, N., Bisson, J. F., Desor, D., Javelot, H., & Rougeot, C. (2011). Beneficial psychological effects of a probiotic formulation (Lactobacillus helveticus R0052 and Bifidobacterium longum R0175) in healthy human volunteers. Gut microbes, 2(4), 256-261.
  4. https://www.ipaeurope.org/press-release-probiotics-in-europe-country-results-of-the-consumer-survey-2/ 
  5. https://www.chr-hansen.com/en/media/press-releases/2022/2/new-study-of-consumer-understanding-of-probiotics-points-to-significant-opportunities-for-the-food
  6. https://www.fortunebusinessinsights.com/industry-reports/probiotics-market-100083 

C.2

While change in microbiota through probiotic supplementation is the obvious mechanism through which this intervention is supposed to work, the authors provide no data on microbiota dynamics in this manuscript. Even without efficacy in their primary or secondary outcome markers there is value in reporting the microbiota dynamics associated with the intervention. Maybe there was an effect on the outcomes in individuals in whom microbiota changes were observed during the trial (in either study arm). Without including data on microbiota dynamics there is little value to this current manuscript.  

  1. We really thank the reviewer's comment and discussion, we totally agree with what he/she is saying. The microbiome is -without a doubt- the perfect candidate to explain what we are observing and a very valuable measurement for going deeper regarding mechanism. Similarly, as in comment 1, the decision of including or not the microbiome data in this report was subjected to deep group discussion during the trial design and then during the data analysis. We finally decided not to include it for the following reasons:
  • The focus and main objective of the present report was to highlight the relevance of including lifestyle data on clinical trials. Controlling for those variables might clarify what is really happening when an expectable effect is not observed in the whole sample. We are convinced that this main message itself might be useful for many types of interventions such as with probiotics, other supplements or even different types of interventions such as mindfulness. We believe that adding more data to this report will hinder this main message, that is actually a gap in the literature (1). It is worth mentioning that the current special issue is focused on both lifestyle and wellbeing, which matches with the actual data included here.

  • There is almost no data including the microbiome, a full battery of psychological assessments, and lifestyle variables in the context of a randomized placebo-controlled clinical trial using probiotics. Reporting these data includes a full description of a) baseline relationships, b) intervention-related effects, and c) follow-up measurements. Such an amount of data deserves its own history. In this regard, we believe that both (the current piece and the microbiome data) are different (yet related) stories that will have a relevant impact in different scientific fields by themselves, one of them from a more clinical perspective and the other one from a more biological/mechanistic one.

  • There are plenty of well-conducted and cited reports describing some probiotic effects using healthy volunteers, and without including microbiome data (2-4). All of those reports have been contributed to literature from a more clinical perspective which is as valuable as those providing mechanistic discussions. We respectfully disagree that more data changes the value of a particular report, and even more, we think that in this particular case, the microbiome data would obscured the clinical contribution we wanted to make with our manuscript. 

  1. https://isappscience.org/can-diet-shape-the-effects-of-probiotics-or-prebiotics/?fbclid=IwAR0DBa8GasPEd7Rm6qMbq_M1KmhDGO4bPeOK528Xd0A92GAnSCePyHLtgyQ
  2. Kelly, J. R., Allen, A. P., Temko, A., Hutch, W., Kennedy, P. J., Farid, N., ... & Dinan, T. G. (2017). Lost in translation? The potential psychobiotic Lactobacillus rhamnosus (JB-1) fails to modulate stress or cognitive performance in healthy male subjects. Brain, behavior, and immunity, 61, 50-59.
  3. Benton, D., Williams, C., & Brown, A. (2007). Impact of consuming a milk drink containing a probiotic on mood and cognition. European journal of clinical nutrition, 61(3), 355-361.
  4. Steenbergen, L., Sellaro, R., van Hemert, S., Bosch, J. A., & Colzato, L. S. (2015). A randomized controlled trial to test the effect of multispecies probiotics on cognitive reactivity to sad mood. Brain, behavior, and immunity, 48, 258-264.

C.3 

There are also some concerns with data presentation. In table 1 the authors provide p-values for the differences between placebo and probiotic groups. This is nonsensical as we know the likelihood for the observed differences between the groups to be due to chance is actually 1, as subjects were randomized in this study (assigned into groups by chance).

We thank the reviewer for the comment, it is an interesting point to discuss. While it's true that the randomization process should make both groups comparable (and the p value equal to 1), that might not be the case for different reasons (for example, one can randomly select the participants with the highest bmi to be on one group, which will pull the group mean towards those values). Since randomization does not neccesarely rule out confounds (1), providing the p-values is a way of showing that our groups are not biased and making the readers be sure that the randomization process works as intended, by providing a statistical measure assessing the difference between the groups. Moreover, we think that providing extra information (p values in this case) does not harm the readability of the manuscript, but provides a numerical argument that might be useful to some readers.

  1. Deaton, A., & Cartwright, N. (2018). Understanding and misunderstanding randomized controlled trials. Social science & medicine, 210, 2-21.

C4. 

In the results section the authors state “the mean BMI was 22.58 ± 3.56 (Placebo: 23.29 ± 3.5; Probiotic: 22.4 ± 3.6)”. With equal sized groups it is unlikely that this data is accurate as the mean BMI should have been close to 22.85” Maybe just a typo, but it’s concerning that none of the authors detected such an obvious error.

We thank the reviewer for noticing this typo. The mean BMI was corrected in the manuscript. The raw manuscript data is available by request here https://osf.io/hsey2/.

Round 2

Reviewer 2 Report

The authors submitted an extensive response, in which they basically provide the reason for their disagreement with this reviewer's comments. While there clearly is value in evaluating the effects of probiotics on mental health, how this can be objectively addressed in a healthy population needs to be described in detail. For instance while lower levels of anxiety might be beneficial they also can reflect detrimental lack of motivation/excitement. Unless the individual suffers from anxiety issues, changes in the range of normal aren't necessarily either detrimental or beneficial. Especially in the context of lack of effect on mental health measures there is a need to report the effects of the probiotic intervention on the gut microbiota composition as the reader needs to know if the intervention resulted in the desired microbiota changes. If not, then there maybe wasn't a sufficient probiotics supplementation. Regarding the inclusion of the p-value in table 1: as there was randomization the reader knows that the likelihood that the differences observed were due to chance are 1; many journals have a clear policy about not including meaningless p-values.

Author Response

Before we start addressing the reviewer’s suggestions and comments, we would like to thank again the reviewer. Addressing the following points has undoubtedly improved the quality of the current manuscript version. We divided the full paragraph in 2 main points that will be clarified in the following response letter.

C.1

While there clearly is value in evaluating the effects of probiotics on mental health, how this can be objectively addressed in a healthy population needs to be described in detail.

 For instance while low levels of anxiety might be beneficial they also can reflect detrimental lack of motivation/excitement. Unless the individual suffers from anxiety issues, changes in the range of normal aren't necessarily either detrimental or beneficial.

  1. We thank the reviewer's comment. We have included a paragraph in the discussion section commenting on the gaps in probiotics and mental health clinical trials that our research addresses. Please see orange highlights on Page 15. On page 19 we included a caution note regarding the implications of changes in anxiety and emotional regulation.

C.2

Especially in the context of lack of effect on mental health measures there is a need to report the effects of the probiotic intervention on the gut microbiota composition as the reader needs to know if the intervention resulted in the desired microbiota changes. If not, then there maybe wasn't a sufficient probiotics supplementation.

  1. We appreciate the reviewer's concern about the relevance of the microbiome data in this piece. As we previously argued in the reviewer’s response 1, we understand the microbiome data might provide information that complements our results and conclusions. However, not including this data is a decision we would like to keep (we already explained the reasons for this decision). We added a paragraph in the limitation section discussing the points the reviewer kindly suggested. Please see orange highlight on page 18
